# Estimation of Excess Mortality and Years of Life Lost to COVID-19 in Norway and Sweden between March and November 2020

**DOI:** 10.3390/ijerph18083913

**Published:** 2021-04-08

**Authors:** Martin Rypdal, Kristoffer Rypdal, Ola Løvsletten, Sigrunn Holbek Sørbye, Elinor Ytterstad, Filippo Maria Bianchi

**Affiliations:** 1Department of Mathematics and Statistics, UiT—The Arctic University of Norway, 9019 Tromsø, Norway; kristoffer.rypdal@uit.no (K.R.); sigrunn.sorbye@uit.no (S.H.S.); elinor.ytterstad@uit.no (E.Y.); filippo.m.bianchi@uit.no (F.M.B.); 2Department of Community Medicine, UiT—The Arctic University of Norway, 9019 Tromsø, Norway; ola.lovsletten@uit.no

**Keywords:** COVID-19, years of life lost, excess mortality, mortality displacement

## Abstract

We estimate the weekly excess all-cause mortality in Norway and Sweden, the years of life lost (YLL) attributed to COVID-19 in Sweden, and the significance of mortality displacement. We computed the expected mortality by taking into account the declining trend and the seasonality in mortality in the two countries over the past 20 years. From the excess mortality in Sweden in 2019/20, we estimated the YLL attributed to COVID-19 using the life expectancy in different age groups. We adjusted this estimate for possible displacement using an auto-regressive model for the year-to-year variations in excess mortality. We found that excess all-cause mortality over the epidemic year, July 2019 to July 2020, was 517 (95%CI = (12, 1074)) in Norway and 4329 [3331, 5325] in Sweden. There were 255 COVID-19 related deaths reported in Norway, and 5741 in Sweden, that year. During the epidemic period of 11 March–11 November, there were 6247 reported COVID-19 deaths and 5517 (4701, 6330) excess deaths in Sweden. We estimated that the number of YLL attributed to COVID-19 in Sweden was 45,850 [13,915, 80,276] without adjusting for mortality displacement and 43,073 (12,160, 85,451) after adjusting for the displacement accounted for by the auto-regressive model. In conclusion, we find good agreement between officially recorded COVID-19 related deaths and all-cause excess deaths in both countries during the first epidemic wave and no significant mortality displacement that can explain those deaths.

## 1. Introduction

There is an ongoing scientific and public debate worldwide about the optimal strategy for mitigating the negative impacts of the COVID-19 pandemic [1,2,3,4,5,6]. In Europe, most countries executed strong non-pharmaceutical interventions in March 2020 to combat the disease’s explosive spread and, by early summer, the epidemic was reasonably controlled. Among the western European countries, Sweden was an exception, adopting a strategy of implementing mainly voluntary measures [7]. As a consequence, the rate of confirmed cases entered a second and more substantial wave in June and a third and even stronger one throughout the autumn, coinciding with the widespread second wave in Europe. In Europe, the COVID-19-specific mortality rate saw one broad wave lasting from March until July, then a calm period from August until October when a second wave started. The confirmed cumulative COVID-19 death toll in Sweden until 11 November was 6247, which corresponds to 611 deaths per million [8]. This figure is typical for Europe but high compared to Sweden’s Nordic neighboring countries. In particular, Norway, which is very similar to Sweden in terms of culture, demographic development, and social policies [9,10], has chosen a much more strict approach against COVID-19. As a result, by 11 November, Norway had only 285 confirmed deaths (53 per million) related to COVID-19 [8].

It has been suggested that the criticism of the Swedish strategy has been based on the perception that death from coronavirus infection somehow is more harmful to society than death from another infection [11]. The implicit assumption behind this suggestion is that the pandemic’s mortality rate was not substantially higher than during previous seasonal influenzas and that all-cause excess mortality in Sweden differed significantly from the confirmed coronavirus-related mortality throughout the pandemic wave. In this paper, we investigate the validity of these assumptions. We also estimate the years of life lost (YYL) in Sweden that can be attributed to its relaxed mitigation strategy.

A standard method for estimating excess mortality is to compute how the number of weekly or monthly deaths in 2020/21 differs as a percentage from the average number of deaths in the same period over the years 2015–2019. This metric is called the P-score, and takes the form of a time series with weekly or monthly resolution. According to Our World in Data [12], it peaked at 9 percent on 29 March in Norway, and at 47 percent in Sweden on 12 April 2020. An overview of the excess mortality computed this way for Germany, Italy, Norway, Sweden, and Switzerland during the first wave of the COVID-19 pandemic is given in an EFTA publication dated 16 December 2020 [13].

There are two obvious caveats of this method. One is that mortality from the seasonal influenza varies considerably from year to year, and hence it is not given that the average of the deaths in the same week over the previous five years is a good estimate of the expected mortality for that time of the year. Another is that mortality exhibits a long-term negative trend. Not taking that trend into account leads to an underestimation of the excess mortality during the pandemic. The long-term trend is taken into account in the Z-score employed in the EuroMoMo model [14] but, as explained in Section 2.3, the seasonal trend is not fully accounted for.

Section 2 presents simple methods for estimating the years of life lost (YLL) and the expected (or baseline) signal for mortality including the long term trend and the seasonal variation based on 20 years of data. This allows us to estimate a more reliable excess mortality during the first pandemic wave and the preceding years. We employ all-cause mortality data up to the end of the first wave, but not for the second wave. The reason for this is that the second wave in most European countries initially was transmitted mainly in the younger age groups, in contrast to the first wave that spread quickly in the older part of the population. Mitigation policies were also considerably more similar in Norway and Sweden during the fall and early winter of 2020/21. Therefore, there is ample reason to believe that an analysis of excess mortality for the second and later waves will give different results from what we present here. This is one of the reasons why Vaclav Smil suggests that it is too early to judge Sweden’s COVID-19 policy [15]. Nevertheless, conclusions cannot be drawn if there remains significant doubt about the actual contribution of the first pandemic wave to the excess mortality and to the years of life lost in the neighboring countries, Norway and Sweden.

## 2. Materials and Methods

### 2.1. Data Sources

Weekly mortality data in Sweden and Norway for the last 20 years were downloaded from Statistics Sweden (SCB) [16] and Statistics Norway (SSB) [17]. The proportion of deaths in 2020 in Sweden by age group and the life expectancy by age group were also obtained from SCB and are reported in Table 1. Data for COVID-19-related deaths were obtained from ourworldindata.org [8].

### 2.2. Estimates of Years of Life Lost

Using data on life expectancy in different age groups in Sweden [16] (Table 1) we simulated the YLL using the model: (1)$$\mathrm{YLL}=X\left[0.10r_{1}+0.30r_{2}+0.35r_{3}+0.25r_{4}\right],$$
where the random variable *X* represents excess mortality, with the estimated distribution for 2019/2020, and the random variables $r_{1},\cdots ,r_{4}$ represent the life expectancies of each age group from Table 1. We assumed the life expectancies to be independent and normally distributed random variables. From the reported statistics on life expectancy, the resulting estimate is YLL = 45,850 (13,915, 80,276).

### 2.3. Estimate of Expected Mortality-Rate Signal

The mortality rate in Scandinavia has a seasonal variation and is higher in the boreal winter [18]. As shown in Figure 1A, the weekly number of all-cause deaths also shows a significant negative linear trend ($p=10^{-15}$ for Norway and $p=10^{-7}$ for Sweden) over the last 20 years.

The expected mortality-rate signal is computed from the average seasonality and the linear trend. We first computed the linear trend in the mortality time series by means of simple linear regression. After subtracting the trend, we computed the expected seasonal variation over a year by averaging the number of deaths in the epidemic year (i.e., the time series that goes from July to July of next year) over those 20 years. By repeating this expected seasonal variation over the 20 years and by adding the linear trend, we obtained the expected mortality-rate time series.

In the following, we will refer to such an expected mortality-rate signal as the baseline signal. Our definition of the baseline is different from that employed in computing the Z-score adopted by the widely used EuroMoMo model [14]. Their seasonal baseline function looks like a sinusoidal function, which does not include the peak of an expected winter influenza. This implies that this expected peak will be recorded as excess deaths. The Z-score is reasonable when the seasonal influenza is the main object of study, but not when this object is a pandemic like COVID-19, where the point is to distinguish the pandemic deaths from those attributed to the seasonal influenza.

### 2.4. Estimate of Excess Mortality Rate

The excess mortality rate for a given week is the weekly mortality rate that week minus the baseline at the time. It can be positive or negative, depending on whether the instantaneous mortality rate that week is above or below the baseline.

The excess mortality is a random process where the value for a given week is given in terms of an estimate for the expected value and a 95% confidence interval (CI). The CI for the estimate of the excess mortality-rate was computed using a Monte–Carlo simulation in which we repeatedly randomized the estimated excess mortality-rate signal without changing its correlation structure. This was achieved by first performing a Fourier transform, then randomizing the phases of the Fourier coefficients, and finally inverting the transform [19]. We added this new realization of the excess mortality random process to the previously estimated expected mortality signal. Finally, we made new estimates of the trend and seasonal variation to obtain new realizations of the expected mortality signal. After this procedure, we obtained a distribution from which we could compute the mean and a CI of the excess mortality rate.

### 2.5. Estimate of Mortality Displacement

Year-to-year variations of all-cause mortality are dominated by variations in seasonal influenza, and we should observe negative correlations between excesses in a given year and the following year (or years). In other words, we should observe this negative correlation in the auto-correlation function (ACF) for the weekly all-cause excess time series. We computed the ACF based on 20 years of weekly excess mortality rate data for Norway and Sweden by using the estimator: (2)$$\mathrm{ACF}\left(\tau \right)=\frac{1}{\left(N-\tau \right)\sigma^{2}}\sum_{t=1}^{N-\tau }\left(x_{t}-x_{t+\tau }\right)$$
where τ is the time lag, μ is the sample mean and $\sigma^{2}$ the sample variance of the weekly excess mortality rate signal of length N=1040 weeks. We estimate the ACF from the annual data, while the error bars are computed by estimating the ACF for 52 different signals with annual resolution. We had 52 samples since there are 52 weeks in a year.

The effect of mortality displacement on excess mortality and YLL is studied by modeling the annual excess mortality as a first-order autoregressive (AR1) process. The justification of this modeling choice is based on the estimated ACF, which indicates the presence of very short memory in the excess mortality process. This is reasonable, since we expect oscillations in all-cause mortality to be compensated quickly in the following year(s). The AR1 models the annual excess mortality $X_{t}$ as: (3)$$X_{t+\Delta t}=\phi X_{t}+\xi_{t}\;$$
where Δt=1 yr and $\xi_{t}$ is a white-noise term.

To find the parameter ϕ in Equation (Equation 3) we used the standard maximum-likelihood estimator. The maximum likelihood estimator is known to be biased for a short time series but for small negative values of ϕ, such a bias is negligible [20]. To estimate a distribution for ϕ, we use a bootstrapping method where we simulated the estimated process and re-estimated the parameter ϕ repeatedly. From such a distribution we extract the mean value and a 95% CI.

## 3. Results

### 3.1. The Expected and Excess Mortality Rate

Figure 1A shows the expected (black line) and recorded (red line) all-cause mortality in Norway and Sweden over the last 20 years with weekly resolution. In Figure 1B,C we plot the expected all-cause mortality rate for Norway and Sweden over the epidemic seasons from 2016/17 up to 2020/21 and the recorded rate up to 11 November 2020.

For both countries, mortality during the winters of 2016/17 and 2017/18 was higher than the baseline, mostly because of stronger-than-normal seasonal influenza [21]. In Sweden, the mortality rate in 2018/19 and 2019/20 was below the baseline until the COVID-19 outbreak in March 2020. Still, after 11 March, it was way above until July and then remained slightly below until November. We estimated the excess mortality rate during the epidemic from 11 March until 11 November as the difference between the observed and expected rate. We compared it to the numbers of weekly reported COVID-19 deaths (Figure 1D,E). The excess all-cause deaths were slightly more numerous than the reported COVID-19 deaths in both countries during the peak of the first epidemic wave.

### 3.2. Excess Mortality and Mortality Displacement

In Figure 2A,B we plot the excess mortality rate over the last four years. The blue lines mark the mean excess rate for each epidemic year (from July until July next year).

For both countries, we observe that the two first years are above baseline. For Norway, the year preceding the pandemic was at the baseline, while during the pandemic year 2019/20, the death number was 517 (−12, 1074), where the numbers in the brackets represent the 95% confidence interval. In Sweden, the pre-pandemic year saw −1596 (−2508, −680) deaths (below baseline), while the pandemic year had an excess number of 4329 (3331, 5325). The 255 reported COVID-19 deaths in Norway is within the confidence interval for the excess estimates, and the 5741 in Sweden is slightly above. For the epidemic period 11 March–11 November, however, Sweden had 6247 reported COVID-19 deaths which is within the confidence interval of the 5517 (4701, 6330) excess deaths for this period. Using the same definition, we estimated the annual excess numbers for the last 20 epidemic years (Table 2 and Figure 2C,D).

From Figure 2B,D one can suspect that a negative excess death rate before the pandemic in Sweden has produced a pool of survivors that potentially could be particularly vulnerable to COVID-19. But the existence of this pool does not imply that it actually contributed more than normal to the COVID-19 deaths. If this displacement mechanism really has played an important rôle in determining the fluctuations of the all-cause excess mortality rate in Scandinavia, it should be observable in its time series and not only in the two successive epidemic years 2018/19 and 2019/20. We explore this effect in Section 3.3.

### 3.3. The Tiny Effect of Mortality Displacement on Excess Deaths and YLL

It has been suggested that low mortality in years of low seasonal influenza will spill over to the following year(s) and give rise to higher than normal mortality, and that this effect explains a substantial part of the excess mortality during the COVID-19 first wave in Sweden [9]. Figure 3 shows the estimated ACF for Norway and Sweden based on 20 years of weekly data (1040 data points). The confidence intervals for each year of time lag are given as error bars in the figures. Only a very weak correlation can be detected on time scales longer than the duration of the peak season for influenza. This suggests that mortality displacement is not generally a major driver of excess mortality fluctuations in Norway and Sweden.

In Section 2.5, we described how to model the mortality time series as an AR1-process and how to estimate the AR1 coefficient ϕ, which can be interpreted as the autocorrelation between the excess mortality between two successive years. In Sweden, we observe a slight anti-correlation in the year-to-year excess mortality given by the estimated AR1 coefficient ϕ=−0.11 (−0.50, 0.30). Hence, it is conceivable that the large excess mortality in 2019/20 may cause a response of negative excess mortality in the coming years.

If *X* is the observed excess mortality in 2019/20, the anticorrelation would lead to a negative excess mortality, ϕX, in 2020/21, a positive one, $\phi^{2}X$, in 2021/22, and so on. This would lead to an adjustment of excess mortality during the first COVID-19 wave due to its effect on the predicted mortality in the subsequent years. According to the AR1 model, the adjusted mortality is: (4)$$X_{\mathrm{adj}}=\left(1+\phi +\phi^{2}\ldots \right)X=\rho X,$$
where ρ=ϕ/(1−ϕ).

The estimated mean of ρ is −0.06, the median is −0.10, and the 95% CI is (−0.33, 0.43). Using the distribution of ρ to take the effect of possible displacement into account, the excess mortality in Sweden for 2019/20 is adjusted to a mean value of 4098 (2706, 6421) (Figure 4A). Carrying out the estimates of YLL with this distribution of excess mortality in 2019/20 we obtained an YLL estimate of 43,073 (12,160, 85,451). Compared to the result in Section 2.2, the mean is reduced by 6% (Figure 4B).

## 4. Discussion and Conclusions

The debate about the necessity of non-pharmaceutical intervention, and the extent of such measures, is still raging, and in Scandinavia one of the hottest issues in this debate is how to measure the real mortality and the years of life lost that can be attributed to COVID-19. The purpose of this paper is to scrutinize claims that all-cause excess mortality deviates substantially from the official COVID-19 mortality reported in Norway and Sweden, and that an explanation for such a claimed difference could be explained as a mortality displacement from one season to the next.

Since COVID-19 mortality is much higher among the elderly and frail, it is also often argued that even though the death numbers are high, the years of life lost may be rather insignificant. For policy makers, who generally have to weigh the perils of YLL against those of strong interventions, it should be of interest to have the YLL quantified. Thus, this has been the second objective of this paper.

A dynamic web page published by The Economist, updated on 9 March 2021 [22], shows a time series of the weekly number of deaths that governments have officially attributed to COVID-19, which are compared to all-cause excess death figures in 71 countries, including Norway and Sweden. The general worldwide picture shown there is that the waves in excess mortality coincide with those in the official COVID-19 mortality, but typically are somewhat higher, indicating that the general under-reporting of COVID-19 deaths is more important than the over-attribution of deaths to COVID-19 due to failure of paying notice to co-morbidity.

Until 14 October 2020, this page applied the P-score for the estimation of excess deaths, but after this date it appears that they have applied the Z-score or a method similar to ours (see Section 2.3). Their graphs for excess deaths and official COVID-19 deaths for the first wave in Norway and Sweden are almost identical to Figure 1 D and E. On that page the curves are continued into the second wave and up to 14 February 2021. In Norway, the excess all-cause mortality drops in the late autumn and goes negative around 1 December, while the COVID-19 death rate remains low and fairly constant. In Sweden the two death rates grow and are almost the same up to 1 January 2021, but then the all-cause rate drops rapidly and becomes negative on 7 February, while the COVID-19 rate peaks at 17 January, and then starts to decline. Up to the end of 2020, the rate COVID-19 deaths per capita stays about 10 times higher in Sweden than in Norway, and so does the all-cause mortality. The drop in all-cause mortality in both countries after new year may be attributed to the total absence of the seasonal influenza so far in the season 2020/21 and perhaps to some extent, the increasing number of elderly immunized by vaccines.

Even though our results confirm those presented by the The Economist, they differ from those of Juul et al. (2020) [9], who suggest that all-cause mortality in Norway and Sweden during the first wave of the COVID-19 epidemic up to July 2020 was largely unchanged compared to the previous four years and that the high excess mortality observed in Sweden during the epidemic wave was partly due to a mild influenza season during the winter of 2019/20. In that paper, the 5741 COVID-19-related deaths in Sweden reported between 11 March and 26 July were interpreted partly as a mortality displacement within the epidemic year 2019/20 and from this year to the next, with the implication that few years of life were lost.

It is commonly claimed, as done in [9], that all-cause mortality rates are more reliable than reported COVID-19-related deaths. The results presented in this paper show that if our model for estimating the expected mortality rate is used, the two rates agree within the confidence range of the estimated all-cause excess rate. Our corresponding estimates of YLL are consistent with Oh et al. [23].

Another central issue raised in [9] is whether the COVID-19 peak in the all-cause mortality rate observed in Swedish data could be explained as mortality displacement, either from the preceding year or from the months preceding the epidemic wave within the epidemic year 2019/20, or from both. In Section 3.2 we found that the negative excess death number (−1596) in 2018/19 constitutes less than 40% of the positive excess (+4329) in 2019/20, so such a displacement can at best only explain part of this excess. Most likely, there is no strong causal link between excess mortality in those two years, since our estimate of the auto-correlation function of the excess mortality over the last 20 years shows a very small negative correlation between successive years. This weak correlation also indicates that mortality displacement makes an insignificant correction to our estimate of more than 40,000 years of life lost in Sweden as an effect of the first pandemic wave.

## Figures and Tables

**Figure 1 ijerph-18-03913-f001:**
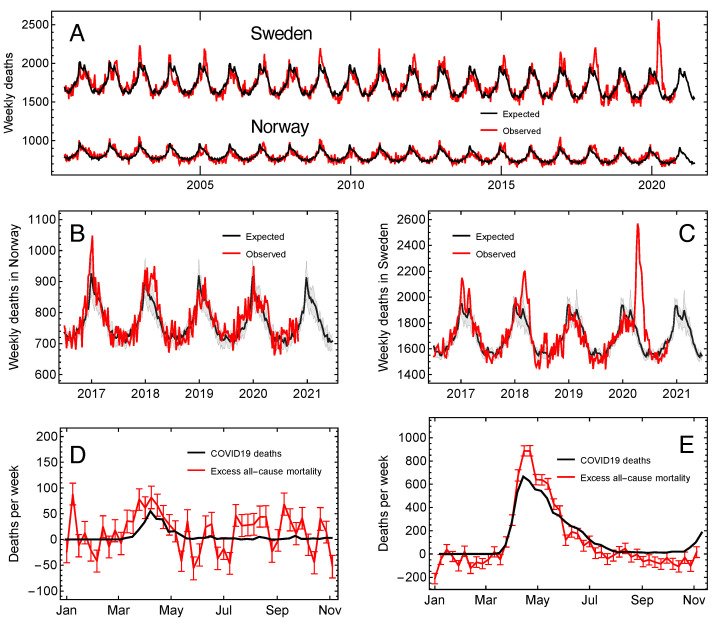
Expected and observed mortality. (**A**) The weekly deaths in Norway and Sweden (red) together with the estimated baseline (black). (**B**) Same as for Norway in (**A**), but for the years 2016/17 to 2020/21. The gray region shows the interquartile range for the seasonal variation. (**C**) As (**B**), but for Sweden. (**D**) The excess weekly mortality in Norway (red) and COVD-19-related deaths (black). The error bars is the 95% CI for the excess mortality based on the Monte–Carlo simulation for the estimate of the baseline. (**E**) As (**D**), but for Sweden.

**Figure 2 ijerph-18-03913-f002:**
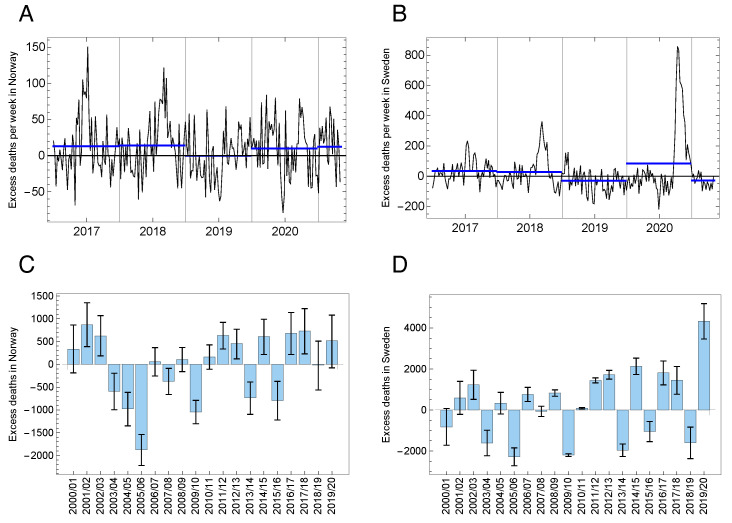
Excess mortality. (**A**) Weekly excess mortality for Norway from 2016/17 and through the first months of the epidemic year 2020/21. The blue lines are the average values for each of the five epidemic years. (**B**) As (**A**), but for Sweden. (**C**) The annual excess mortality for Norway from 2000/01 to 2019/20. The error bars are the 95% confidence intervals. (**D**) As (**C**), but for Sweden.

**Figure 3 ijerph-18-03913-f003:**
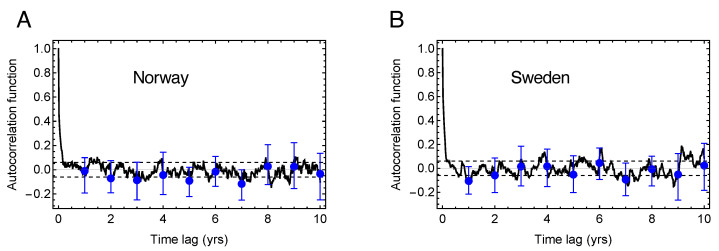
Auto-correlation function of the excess mortality signal. (**A**) The black curve shows the autocorrelation function estimated from the weekly excess mortality in Norway. The dashed lines indicate the 95% confidence interval under the assumption of uncorrelated data. The blue points show the autocorrelation function estimated from yearly excess mortality, and the blue error bars show the spread between the correlations estimated using different weeks of each year. (**B**) As in (**A**), but for Sweden.

**Figure 4 ijerph-18-03913-f004:**
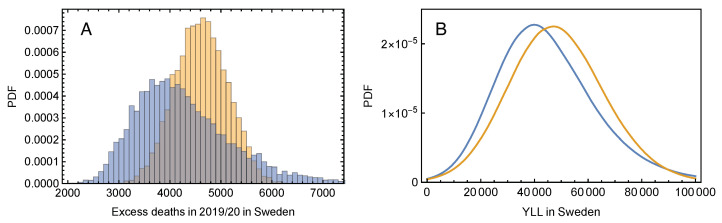
Effect of mortality displacement. (**A**) The yellow histogram shows the estimated probability density function for excess deaths obtained from the Monte–Carlo simulation of the baseline. The blue histogram is the excess mortality adjusted for displacement according to Equation (Equation 4). (**B**) The blue curve is the estimated probability density function for years of life lost (YLL) obtained from Equation (Equation 1), and the blue curve is the probability density function for YLL after adjusting for mortality displacement.

**Table 1 ijerph-18-03913-t001:** Proportion of deaths in 2020 in Sweden by age group and life expectancy by age group. Data source: Statistics Sweden (SCB).

Age Group (yrs)	Proportion of 2020 Deaths	Life Expectancy (yrs) Estimate (SD)
50–64	10%	27.5(3.8)
65–79	30%	15.6(3.3)
80–89	35%	7.0(1.6)
>90	25%	2.5(0.9)

**Table 2 ijerph-18-03913-t002:** Excess mortality per (epidemic) year. The excess mortality is defined as the registered deaths per year minus the expected number of deaths. The expected number of deaths are obtained from a model with a linear trend superposed on a seasonal signal. The confidence intervals are obtained by repeated re-estimates of the linear trend and seasonal signal in a Monte–Carlo simulation. The asterisk in the last row indicates that epidemic year is not completed yet.

**Year**	**Excess Mortality in Norway**		**Excess Mortality in Sweden**	
	**Estimate**	**(95% CI)**	**Estimate**	**(95% CI)**
2000/01	334	(−180,838)	−825	(−1752,84)
2001/02	866	(391,1331)	587	(−261,1410)
2002/03	621	(173,1050)	1227	(466,1946)
2003/04	−591	(−1002,−192)	−1609	(−2281,−956)
2004/05	−977	(−1353,−606)	331	(−261,903)
2005/06	−1874	(−2215,−1527)	−2283	(−2803,−1790)
2006/07	59	(−254,373)	758	(314,1188)
2007/08	−371	(−661,−95)	−67	(−449,305)
2008/09	105	(−161,367)	825	(497,1151)
2009/10	−1043	(−1298,−783)	−2197	(−2505,−1885)
2010/11	163	(−98,435)	87	(−241,422)
2011/12	633	(362,933)	1443	(1073,1825)
2012/13	456	(160,777)	1718	(1281,2156)
2013/14	−732	(−1048,−390)	−1959	(−2467,−1448)
2014/15	608	(254,980)	2131	(1547,2717)
2015/16	−793	(−1178,−390)	−1046	(−1707,−378)
2016/17	682	(258,1111)	1811	(1069,2564)
2017/18	731	(281,1196)	1450	(623,2283)
2018/19	−15	(−516,495)	−1596	(−2508,−680)
2019/20	517	(−12,1074)	4329	(3331,5325)
2020/21${}^{*}$	646	(362,957)	−1501	(−1917,−1079)

## Data Availability

Data and material from this study can be requested by contacting Martin Rypdal (martin.rypdal@uit.no). The analyzed data is also publicly available from Statistics Sweden [16], Statistics Norway [17], and Our World in Data [8,12].

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
