# Peer review of "Estimation of Excess Mortality and Years of Life Lost to COVID-19 in Norway and Sweden between March and November 2020"

_ijerph, 2021, doi:10.3390/ijerph18083913_

Round 1

Reviewer 1 Report

Dear Authors,

Thanks addressing my previous comments and suggestions. I have no further comments. Best of luck with your future research!

Author Response

Thank you for this comment. 

Reviewer 2 Report

First of all, thank you for your effort to improve the quality of manuscript based on reviewers' suggestive comments.

I would like to share your research findings with other colleagues in the forthcoming issue of the prestigious MDPI journal.

After all, I'd recommend this paper deserves to be accepted for making a publication in the journal, IJERPH.

Author Response

Thank you for this comment. 

Reviewer 3 Report

Introduction: The last paragraph of the introduction is unnecessary and does not serve any purpose - can be deleted. The title and subtitles of section 2 &3 are sufficient for the readers to know and prepare to read for the content that follows. It also makes the introduction section longer than needed.

Line 114: is there a typing error 'EuroMMOMO' because the way it is written in the reference provided by the authors is 'EuroMoMo'. Please maintain consistency of terminology. 

Author Response

Thanks for good comments. We have deleted the last paragraph in the introduction and corrected the error in "EuroMMOMO". 

This manuscript is a resubmission of an earlier submission. The following is a list of the peer review reports and author responses from that submission.

Round 1

Reviewer 1 Report

This is article needs to be written in form of a commentary rather than an original article. The authors herein have conducted this research to compare/contradict the results of the study by Juul et.al. 2020 using a different methodology than theirs. However, the paper by Juul et.al. is a preprint and have not yet been published or peer-reviewed. Therefore as a researcher I do not see the need to use Juul's paper as a comparison to conduct this research. The content presented by the authors is valuable but the way it is presented is not fathomable. I request the authors to check the order of the manuscript before submitting for publication in a scientific journal. I have been reviewing manuscripts for quite a few years now, but have not come across any manuscript where the results and discussion section follow the introduction and manuscript ends with the material and methods section. The manuscript cannot be reviewed in the form it has been submitted by the authors. 

Reviewer 2 Report

It is an informative case study of the COVID-19 in Sweden. I would like to share your research findings with other colleagues later on. However, the title of manuscript is required to be revamped in clear and to the point. Please make an elaboration of conclusions, and some recommendations or suggestions based on the study outcomes. Also, key benefits or contributions of your work should be addressed as well. Therefore, in order to improve the quality of paper, and eventually make a successful publication in the forthcoming issue of IJERPH, the article has to be further revised in accordance with reviewer's constructive comments. After all, this manuscript would be accepted on condition of major amendment.

Reviewer 3 Report

Dear Authors,

Thanks for the opportunity to review this paper. I found it interesting and relevant. Below are, in my view, some suggestions on how to further develop this paper. Please consider them as suggestions.

Abstract

To enhance reader friendliness, I would suggest, in text, to more clearly define (and spell out) the overall aim and purpose. Also, I miss some information about the study`s theoretical and practical implications and recommendations.

Introduction

  • Line 21: “a more relaxed approach” does not seem very accurate or academic. Consider using different wording.
  • Line 28: In what ways are Norway and Sweden alike? A couple of examples here would strengthen the argument.
  • Line 32: This argument is a little hard to grasp/understand. It might be a language issue. Consider re-writing for clarity
  • Line 39: Good summary of the paper`s aim. I would also suggest adding an outline of the rest of the paper
  • Line 40: The differences in results compared to Juul et al. (2020) is interesting. However, I would recommend moving that to the discussion section

Theoretical framework?

  • I do miss a section related to this paper`s throetical framework. The above introduction presents the background in a good way, but what about theory? A theory section needs to be included

Results

  • This section appears as a mix between methods, findings, and results. This makes it hard to follow. It might be the template, but if not, I would suggest moving section 4 up before the results section, and structure the paper: abstract - introduction – methods – findings- discussion – conclusion
  • Limitations? This is missing.

Discussion

  • This section is too short and needs more work. The purpose of this section is to interpret and describe the significance of your findings in light of earlier discussed academic literature (which, in my view, is missing).
  • This section seems more as a result section than a discussion. I would recommend moving the table up to the result section and instead, in this section, focus more on interpretation and discussion.

Conclusion

This section is, in my view, missing. I would recommend adding a section where you outline your main findings, talk about implications and make recommendations for further research